# The Role of Sport Psychology in Injury Prevention and Rehabilitation in Junior Athletes

**DOI:** 10.3390/bs14030254

**Published:** 2024-03-20

**Authors:** Moritz Weiß, Matthias Büttner, Fabio Richlan

**Affiliations:** Centre for Cognitive Neuroscience & Department of Psychology, Paris-Lodron-University of Salzburg, Hellbrunnerstr. 34, 5020 Salzburg, Austria; moweiss200@gmail.com (M.W.); matthias.buettner@stud.plus.ac.at (M.B.)

**Keywords:** athletes, injury, pain, prevention, psychology, rehabilitation, sport

## Abstract

Sports injuries have historically been addressed and treated from a purely physical perspective. Nevertheless, like in many other aspects of sports, it has become evident during the last decades that psychological considerations and consequent interventions are both vital and inevitable in the work with athletes, particularly in the work with junior athletes. Especially in the domains of sports injury prevention and rehabilitation, psychological measures can yield significant benefits for junior athletes. Stress management techniques, cognitive restructuring, mindfulness, motor imagery, or seeking social support have been demonstrated as being highly effective. These techniques, many of them originally intended by sport psychologists to optimize performance, now aid junior athletes in performing at their best while also preventing injury and facilitating a safe return to competition after injury. During injury rehabilitation, sport psychological measures play an important role as well. The purpose of this review is firstly to provide an overview of the psychological factors that significantly support both injury prevention and rehabilitation. We subsequently elaborate on the identification and optimization of these factors by presenting evidence-based psychological interventions and training programs. In addition, we provide science-informed fundamentals that may serve as a basis for the adaptation and/or development of novel psychological measures to support junior athletes during injury prevention and rehabilitation.

## 1. Introduction

Sports injuries have historically been addressed and treated from a purely physical perspective. With significant progress in the medical field, the advent of new knowledge, and technological advancements, athletes can now return to competition following injury faster and safer than ever before. Nevertheless, like in many other aspects of sports, it has become evident during the last decades that psychological considerations and consequent interventions are both vital and inevitable in the work with athletes, particularly in the work with junior athletes. Especially in the domains of sports injury prevention and rehabilitation, psychological measures can yield significant benefits for both elite and junior athletes [1,2,3,4,5,6].

Objective evidence suggests that psychological interventions can substantially lower injury risk [7,8,9]. Notably, stress management techniques, such as breathing exercises, cognitive restructuring, mindfulness, motor imagery, or seeking social support, have been demonstrated as being highly effective [7,10,11,12]. These techniques, many of them originally intended by sport psychologists to optimize performance in elite athletes, now aid junior athletes in performing at their best while also preventing injury and facilitating a safe return to competition after injury, if indicated.

Despite these and other preventative measures, injuries unfortunately remain an existential and inherent threat in the lives of athletes [13]. During injury rehabilitation, sport psychological measures play an important role as well. For instance, if junior athletes respond extremely negatively to an injury and encounter significantly more negative emotions and thoughts than they did prior to injury, this can have devastating impacts on the progress of rehabilitation [14]. The resulting stress leads to poorer recovery via reduced sleep quality, weakened immune system, and deficient cognitions [15], among other mechanisms, which, in turn, often result in harmful behaviors such as avoiding rehabilitation exercises or withdrawing from social connections [16,17].

The purpose of this review is firstly to provide an overview of the psychological factors that significantly support both injury prevention and rehabilitation in junior athletes. We subsequently elaborate on the identification, optimization, and individual variability of these factors, when necessary, by presenting evidence-based psychological interventions and training programs. In addition, we provide science-informed fundamentals that may serve as a basis for the adaptation and/or development of novel, individually tailored psychological measures to support junior athletes.

## 2. Factors and Predictors

Wiese-Bjornstal et al. [18] incorporated Williams and Andersen’s [19] stress-injury model into their response to a sports injury and rehabilitation model. The model distinguishes between personal and situational factors that affect the course and outcome of rehabilitation.

A key component of the model is cognitive appraisal, which involves a psychological assessment of the injury itself and one’s ability to cope with it. This evaluation can impact the rehabilitation progress positively or negatively, via affecting athletes’ emotions and behaviors during the process. A positive assessment may involve viewing the injury as a hurdle to be overcome on one’s personal journey. This mindset allows setbacks to be treated as temporary obstacles on the path to growth. Additionally, having a strong sense of self-efficacy can be beneficial.

In contrast, in unfavorable evaluations the injury is interpreted as a potential threat or setback, which, in turn, results in undesirable emotional states and behaviors. If junior athletes feel helpless due to their injury, they might abruptly discontinue their training regimen or execute rehabilitation exercises without the necessary commitment. If the targeted goals subsequently have to be modified or delayed, athletes may become further entrenched in their unfavorable evaluation, thereby emotionally spiraling downwards [20]. Unfortunately, appraisal of the situation as a threat or loss is particularly likely in sports injuries [21].

### 2.1. Situational Factors

Situational factors can be broadly divided into three categories: sport, social environment (relationships), and general environment.

Sport includes the type of sport played, the level of competition, and the current situation in the season, all of which can impact rehabilitation. For instance, the season’s circumstances may prompt athletes to return to competition sooner than advisable to contest for a title or stay at a certain level of competition.

The social environment encompasses the impact of teammates, coaches, friends, family, and relatives on junior athletes and therefore their rehabilitation process. Furthermore, the support offered by clubs/federations and their approach towards injury management can have a substantial influence on recovery outcomes. This influence is described in more detail in Section 4 (stress management training and interventions).

Tranaeus et al. [8] highlight the importance of the professional relationship between athletes and sports physiotherapists or other medical professionals. These individuals are in a prime position to offer social support and guide athletes’ emotions and behaviors due to their constant contact and crucial role in rehabilitation. Objective and concise communication between athletes and medical professionals is essential for successful rehabilitation. In the case of junior athletes, this professional relationship is even more important, due to their limited knowledge or (mis)beliefs, causing a greater dependence on physiotherapists. In this context, the important role of parents is also worth mentioning, since their influence is particularly strong in junior athletes compared to adult athletes.

Finally, the overall environment should also be considered as a relevant factor in the rehabilitation process. The success of rehabilitation is strongly influenced by the availability of specialists, physiotherapists, and other professionals, as well as the necessary equipment to implement prescribed measures.

### 2.2. Personal Factors

Stress is a significant factor in the life of athletes, affecting performance both in desirable ways (i.e., performance increase) and undesirable ways (i.e., performance reduction) (e.g., Fuchs and Gerber [22]). Up to this point in this article, the term “stress” was generally associated with negativity and has been discussed as a condition that should be minimized and controlled. Negative consequences, however, are mostly associated with experiencing strong, acute, or chronic stress (for a review see Yaribeygi et al. [23]). Moderate stress or stress that occurs only in confined performance situations can actually have positive effects.

Activation through a moderate level of stress releases performance reserves that are otherwise not available [24]. Arent and Landers [25] found that participants performed a simple response time task best at 60–70% of maximum arousal. The study further suggests that the inverted U-model of arousal and performance is able to accurately predict results. Thus, moderate stress can lead to an increase in performance. Additionally, dealing with stressful experiences can help in developing coping strategies for stress that can be applied to similar situations in the future, and increase the perceived confidence of junior athletes.

The rehabilitation process, however, largely depends on the injury itself. Specifically, the outcomes of rehabilitation are influenced by the severity, type, and injury history. The athletes’ expectation and subjective perception of the success of the rehabilitation significantly affects the course and outcome of the process [26].

## 3. Diagnosis of Stress

To identify the source of stress and to detect early negative developments, sport psychologists often use subjective self-report questionnaires. These questionnaires come in various forms and may be specifically designed for a particular domain. Once the origin and severity of the stress have been determined, the athletes can be trained in techniques and tools to alleviate the negative impact of stress.

A frequent challenge faced by junior competitive athletes is the double burden of sports and school [27]. In line with this double burden, Hamlin et al. [28] observed that stress levels in university athletes were highest during pre-season and during the examination period. This was attributed to the increases in athletic and academic workload, respectively.

Furthermore, due to their usually extremely high drive and motivation, junior elite athletes are already very susceptible to under-recovery. If prolonged, this may necessitate additional rest and seeking professional medical and or psychological care [29]. Moreover, insufficient recovery can heighten the risk of injury and burnout, which could result from overtraining, chronic under-recovery, or other factors. Examples of such other factors that promote injury and/or burnout include social pressure, a one-dimensional identity as a competitive athlete, and a self-esteem that is highly dependent on success in sports [30]. Another vulnerability factor is a lack of coping skills, or maladaptive coping mechanisms [31].

### 3.1. Diagnosis of Stress in Childhood and Adolescence

Several diagnostic self-report questionnaires are available to measure stress levels in children and adolescents [24]. The Perceived Stress Scale (PSS) by Cohen et al. [32] assesses current stress and the resilience towards stressors in adolescents that are at least 12 years old. There are three versions of the PSS: the original PSS-14, the PSS-10, and the PSS-4. The PSS-10 has shown superior psychometric properties compared to both other versions [33,34]. The PSS-10 includes only 10 questions and can therefore be administered and evaluated in just a few minutes. Another example is the Questionnaire on Resources in Childhood and Adolescence (FRKJ 8-16) by Lohaus and Nussbeck [35]. The FRKJ aims to identify strengths and weaknesses in coping with stress and promote long-term resources appropriately. The questionnaire captures personal resources such as self-efficacy, self-control, empathy, and perspective taking, as well as social resources like peer group integration and parental emotional and social support. The FRKJ assessment takes 20–30 min to complete and 5–10 min to evaluate. An evaluation of 370 adolescents showed that the FRKJ can differentiate between adolescents with and without psychosocial deficits [36].

While the FRKJ identifies current competencies and resources, it is recommended to additionally evaluate the existence of psychological and psychosocial deficits. This can assist in identifying junior athletes who may benefit from stress management training the most. For this purpose, the Strengths and Difficulties Questionnaire (SDQ) is an appropriate tool, particularly due to its comparatively shorter completion time (3–10 min) in contrast to other alternatives [24], without compromising quality. The SDQ is available with both self-report and other-report versions, which allows for the inclusion of the athletes’ social environment. Below are some examples of self-report items of the SDQ designed for 11–17-year-olds:

“I usually do as I am told”, “I worry a lot”, “I think before I do things”, and “I am nervous in new situations. I easily lose confidence”.

### 3.2. Stress Management Training and Interventions

In this section, we present scientifically tested stress management training tools and provide recommendations on how to best support junior athletes in coping with stress. The ultimate objective of all interventions is to support the development of skills that foster healthy stress management, thereby contributing to athletic growth and injury prevention [7].

Sallen [37] offers a variety of stress management programs tailored for competitive athletes. These stress management methods, which include techniques commonly used in psychotherapy such as cognitive restructuring and relaxation techniques, were found to be beneficial by both the authors and the athletes themselves. Examples of stress management training for competitive athletes include trainings by Krohne and Hindel [38] and Steffgen [39], which were successfully evaluated in table tennis players.

Similar training programs for long-distance runners [40,41] and triathletes [42] also exist. More recent developments aim to promote mindfulness, attention, and acceptance [43,44], which are well-established in psychotherapy; for example, acceptance and commitment training (or therapy, a.k.a. ACT) [45,46,47].

One mindfulness-based approach that has received a lot of attention in recent years is Mindfulness-Based Stress Reduction (MBSR) [48,49]. MBSR has been shown to improve key aspects of mental well-being such as mood and anxiety in both healthy [50] and injured athletes [51] as well as increase pain tolerance in injured athletes. Another well-known mindfulness-based approach is Mindfulness–Acceptance–Commitment (MAC). Zadeh et al. [52] used a modified version of the MAC protocol by Gardner and Moore [53] on 23 semi-professional soccer players who subsequently showed significantly lower injury rates than a control group. It should be noted that none of these trainings were specifically developed for children and adolescents. Therefore, intervention strategies for this target group must be tailored accordingly.

It is also crucial to consider the variability in individual athlete responses to these psychological interventions. Findings by Moesch et al. [54] emphasize this individual variability. They conducted a case study on six severely injured athletes using a mindfulness- and acceptance-based intervention. While all athletes showed some improvements on scales such as acceptance and well-being, the number of scales that athletes improved on as well as the magnitude of the improvements varied considerably.

Fuchs and Gerber [55] provide explicit recommendations for effective stress management in elite sports, including the following.
Developing an understanding of the concrete factors that can trigger stress in competitive athletes across varying career phases, with particular consideration given to junior athletes who may face the added stress of balancing sport and school obligations.Avoiding under-recovery and overtraining and ensuring that junior athletes are adequately recovering, particularly during intensive training and competition phases. To achieve this, it is crucial to identify symptoms of overtraining promptly and monitor the athletes’ recovery–stress balance [56].Those involved in competitive sports, including team coaches, physiotherapists, and sport psychologists, must be aware of the symptoms and triggers of burnout to create a training environment that reduces the likelihood of burnout.These sports professionals should be familiarized with common stress-resistance training techniques for both elite and junior athletes so they can teach missing stress management skills if needed.

### 3.3. Fundamentals for Measures to Improve Stress Management

Lohaus [24] utilizes Lazarus and Folkman’s [57] transactional stress model as a foundation for stress management training for children and adolescents. The model posits that the appraisal of a situation comprises two phases; the amount of resulting stress hinges on the outcomes of these appraisals. In the initial phase, the demand of the situation is evaluated as either irrelevant/positive or unfavorable/difficult, where the two latter evaluations trigger stress. The second phase involves the assessment of available coping resources. Inadequate coping resources entail the occurrence of stress. Lohaus suggests several starting points for interventions based on this model.

Initially, athletes should heighten their sensitivity to potential stressors and recognize what triggers stress in them. This allows for the deliberate and specific implementation of coping mechanisms in the future.

A second strategy involves engaging in cognitive restructuring to reclassify situations as less threatening. The objective is to label situations either as irrelevant/positive (meaning no stress is elicited at all) or as challenging (which induces stress, but to a much lesser extent than the situation being categorized as threatening). Here, training strategies are utilized to assess different situations in the context of competitive sports. The focus is specifically on positive or challenging aspects of the situation. Lohaus emphasizes that exercises promoting self-esteem are additionally beneficial, as this creates a more positive outlook on many situations.

The third and fourth starting points relate to a secondary assessment, which evaluates whether an individual’s coping resources are adequate for the situation. Stress primarily arises when coping resources are evaluated as insufficient. Gomes et al. [58] found that coping appraisal partially mediates the effects of work-related stressors on psychological health. The third point deals with enhancing coping skills, specifically teaching junior athletes new techniques or strategies (see Section 3.4 and Section 3.5). It is best to train newly acquired resources directly by applying them to specific situations from the athletes’ lives or situations that they are likely to encounter in the near future. Thus, the fourth starting point refers to the training of coping strategies.

The fifth and final point involves identifying the symptoms and consequences of stress at an early stage. This is essential for implementing prompt stress reduction measures and because stress reactions can further trigger additional stress. Purcell et al. [59] recommend regular mental health screenings for athletes, particularly during periods of risk such as after a severe injury or both before and after major competitions.

### 3.4. Examples of Stress Management Training for Children and Adolescents

This section will examine the EPHECT (Effects of a Physical Education-Based Coping Training) program as an instance of stress management training that is designed for children and adolescents [60,61].

The program is uniquely tailored to be incorporated into sports activities and focuses on developing coping strategies. Specific versions of the program already exist for seven sports (soccer, swimming, judo, gymnastics, dancing, roller sports, juggling); adapting it to other sports should also be straightforward.

An advantage of this approach, particularly for junior athletes, is that they can apply and reflect on newly learned strategies directly within a sporting context. The EPHECT program comprises the following six modules best conducted at weekly intervals.
Understanding and experiencing stress.Successful time management.Developing mental strength.Gaining control over emotions.Eliminating stressors.Tackling stress together.

The program triggers stress reactions in participants through various exercises. By consciously recognizing and reflecting on individual stress reactions, athletes are sensitized towards their own stress reaction as recommended by Lohaus [24]. As this program is conducted in a group setting, it reveals varying degrees of individual stress responses during different exercises, providing athletes with essential information on the situations that evoke high stress levels.

Each module lasts only 20 min to optimize training time utilization. The EPHECT program includes a workbook to supplement the lessons, comprising informative content and brief homework tasks to accomplish following each session. The homework emphasizes the implementation of acquired stress management techniques in various domains of life, specifically education. Thus, the program is aptly designed to tackle athletic and academic pressures simultaneously.

When evaluating the EPHECT program, Lang et al. [60] observed an increase in the use of adaptive coping instead of maladaptive coping, along with students in the experimental group rating their emotion-focused coping skills higher than students in the control group. A similar effect was found between students in the experimental group, with students whose homework was regularly checked reporting more adaptive coping and higher self-reported ratings in problem-focused coping skills. This nuance indicates that reflecting the deliberately induced stress reactions is crucial to the program’s success.

### 3.5. Interventions during Rehabilitation

The stress management techniques mentioned up to this point in this article primarily serve the purpose of preventing injuries in youth competitive sports. Nevertheless, injuries may still happen, with prolonged, severe stress potentially increasing the likelihood of injury through various mechanisms [13,62].

Developing concrete, individually tailored strategies for stress management may not only help to prevent injuries, but also contribute to a more favorable outcome during the rehabilitation phase [14,63,64,65]. If junior athletes have already become proficient in relaxation techniques and regularly seek social support, they can direct their attention exclusively to rehabilitation without needing to acquire stress management skills. Additionally, an injury can more easily be viewed as a challenge on their path to success if athletes have already practiced focusing on positive and controllable factors in different situations.

The individuals’ reactions to an injury (and to subsequent psychological interventions) can vary to a large degree and depend on various factors. From a psychological perspective, the focus is on how junior athletes evaluate the injury and the chance for rehabilitation [14,16,17]. This evaluation, which encompasses the situation and its outcomes, as well as the athletes’ self-efficacy to handle the situation and outcomes, significantly affects the thoughts, emotions, and actions displayed by athletes during their rehabilitation process [5,6,65]. Wiese-Bjornstal et al. [18] propose a model that illustrates the intricate interplay between internal and external factors, which is described below.

If junior athletes possess a proven and proficient set of coping mechanisms for dealing with stress, they may view the rehabilitation process as manageable. In the absence of such techniques, athletes should be provided tools that can be utilized immediately (e.g., in the form of stories) [66].

A good starting point is the use of relaxation techniques that junior athletes can employ when they feel overwhelmed or powerless. They can also use these techniques when needing to spare injured regions while experiencing the urge to move. These techniques may also alleviate feelings of pain. Proven, effective relaxation techniques include progressive muscle relaxation (PMR) and various breathing exercises. These techniques must be adapted based on the nature of the injury, e.g., to avoid tensing injured muscles during PMR.

A significant portion of rehabilitation involves establishing and adjusting goals, as noted in Section 2. Junior athletes should be presented with various goal-setting techniques and options and be informed of their relevance to the rehabilitation process [20]. The emphasis should be on formulating individually relevant goals targeting controllable factors and considering goals as inherently flexible, to prevent frustration or similar issues that may arise from necessary, unexpected adjustments. Additionally, Daumiller et al. [67] found that mastery approach goals were negatively associated with burnout levels and psychosomatic stress, whereas mastery avoidance goals were positively associated with burnout levels. This suggests that goals should be aimed at improving skills and knowledge rather than preventing their loss or stagnation.

As previously noted in the context of injury prevention, social support appears to be the most well-evidenced mechanism for managing stress related to sports injuries [11]. To benefit from social support requires developing a network of friends, family, team members, and intentionally seeking social support. Social support helps one to learn that they are valued and supported regardless of athletic performance or the identity as a competitive athlete (self-worth). For junior athletes, in particular, sports make up a significant aspect of their self-identity. An injury can therefore evoke particularly strong negative emotions and thoughts.

Another effective psychological technique that is especially beneficial for injured junior athletes, is visualization or (motor) imagery training [12]. Focused visualization that engages multiple senses (sight, hearing, smell, touch, taste), such as imagining movements and exercises from sports, or competition scenarios, can alleviate stress caused by prolonged absence from sports. This technique can be utilized—depending on the phase of rehabilitation [6,68]—for working on the execution of specific technical movement patterns, nervousness before/during competition, or targeted comebacks.

As with all other types of psychological interventions during rehabilitation, it is important to tailor visualization protocols to the athletes’ individual needs. This was shown in a case study of an injured Olympic athlete, who reported that his previous rehabilitation experience enabled him to have a more positive mindset and make better use of imagery techniques [69]. The athlete also reported using different types of imagery during different stages of rehabilitation, further highlighting the need to make individual adjustments to interventions during the rehabilitation process.

In addition to visualization protocols, virtual reality technology provides promising opportunities for reducing stress and enhancing sports skills, even without active participation, by means of realistically simulating sports situations in virtual reality [70,71,72]. In sum, the available possibilities in terms of psychological interventions allow for the selection and tailoring of specific interventions to satisfy individual needs and preferences and to account for individual athlete responses to such interventions.

### 3.6. Dealing with Pain

During both training and competition, it is essential to distinguish between pain that can be disregarded and that dissipates after brief recovery periods (e.g., exertion/exhaustion pain), and pain that necessitates refraining from sport (e.g., pain due to injury) [73,74]. Pain plays a crucial role in preventing and rehabilitating sports injuries [5,75].

Recognizing and communicating pain is crucial for all athletes, but especially for junior athletes, as they often conceal their discomfort from coaches and teammates [76]. The potential for such behavior is particularly high before major competitions, which may impact the future professional prospects of junior athletes.

Another factor that contributes to ignoring or masking pain is the prevailing culture and values within the particular sporting community [77,78,79,80,81]. Exerting maximum effort despite discomfort is often viewed as admirable and reinforced by social norms. Pushing through pain and injury may be falsely viewed and heroic or especially tough. Sometimes, this behavior may be even expected by teammates, coaches, fans, and sometimes even parents alike.

The athlete–coach relationship plays a crucial role in determining the impact of such pain-masking behavior. Additionally, attentively observing movement patterns is vital to identifying discrepancies from typical routines, such as favoring a particular body part, at an early stage [82].

The cognitive assessment of pain in junior athletes may lead to suboptimal conclusions. This stems from the fact that the prefrontal cortex is responsible for complex action planning and behavioral control, among other functions, and is not yet fully developed in this age group [83,84]. Children and adolescents may struggle to assess the long-term consequences of their actions and incorporate them into their behavior, underscoring the importance of openly communicating about pain.

It remains crucial to consider pain in rehabilitation when making decisions about future courses of action and modifying objectives accordingly. To facilitate ongoing pain evaluation, a pain diary or questionnaire should be completed on a regular basis. During rehabilitation, it can be assumed that junior athletes are much less inclined to conceal their actual pain levels, which allows for the use of self-report methods. The measurement of pain is highly subjective and must be approached in an individualized manner.

Distraction and relaxation techniques can serve as initial methods to manage pain, as they allow junior athletes to push through discomfort and avoid negative goal setting. Distraction or dissociation can be a useful strategy, particularly in the acute phase after a sports injury [74].

In cases where distraction is not feasible or helpful to alleviate sports injury pain, consciously focusing on the pain can aid in accepting and managing it, safeguarding against negative self-assessment [74]. The focus is on acknowledging the pain and determining the appropriate actions to manage both the pain and the injury. Such emotional reactions are common at the beginning of rehabilitation and can result in additional stress and unwanted behaviors. This highlighting of the pain helps with acknowledging injury-related performance decline and determining the appropriate actions to manage both the pain and the injury. This approach prioritizes addressing the direct cause of the discomfort, rather than addressing potentially overpowering emotional responses such as frustration or sadness. Especially during the following stages of rehabilitation, it may be beneficial to implement acceptance or association as coping mechanisms for pain in junior athletes via storytelling [66].

Furthermore, the presence of others, such as coaches, can affect pain perception. In Lord and Kozar’s [85] study, athletes reported lower pain levels and increased pain tolerance when evaluated by trainers. Therefore, to obtain precise pain level measurements and performance status evaluations, it is preferable for impartial individuals to conduct the assessments.

It is also important to consider how injured junior athletes are treated by their social environment. The reaction of the environment often serves as a model for one’s own interpretation or evaluation of the situation. This tendency can lead to the belief that rehabilitation training is more demanding and effortful than it truly is [86]. Individuals of the same sex and age are especially often used as a reference point. Therefore, it is crucial during the rehabilitation process for all team members and those in close proximity to the injured athletes to be conscious of their behavior. These psychological factors complement pharmacological or physical interventions, such as massage, exercise, and electrophysiological measures.

### 3.7. Readiness to Return

Readiness to return to sport cannot be clearly determined from a psychological perspective and is highly dependent on the individual athletes [6,87]. According to the review by [88], two main psychological characteristics should be considered when deciding whether to return to training and/or competition.

Firstly, there should be minimal psychological barriers, such as fear or anxiety. Fear of reinjury is frequently reported by injured athletes as a significant deterrent from returning to their sport at either their former level or from returning to their sport at all [89,90]. Another significant reason for heightened anxiety and delayed return is insecurity about one’s level of performance [87].

Secondly, it is important to cultivate beneficial resources such as self-confidence and motivation during rehabilitation. Usually, self-awareness and confidence in the injured body region increase while anxiety decreases. It is crucial to note if motivation is intrinsic (from within) or extrinsic (from outside). High levels of intrinsic motivation generally serve as an indicator that athletes are prepared to return to sport [91].

As previously discussed, the presence of these factors and the psychological feasibility of a return to sport have to be determined individually. Since the decision to return is influenced by numerous external factors, particularly in the case of competitive junior athletes (e.g., phase in the season, pressure from coaches, sponsors, and media), honest and open communication with athletes is crucial to prevent conflicts and mitigate additional stress caused by external pressures.

### 3.8. Future Developments and Perspectives

The field of sport psychology is rapidly growing with many new developments in the last few years. Consequently, even more relevant contributions are expected in the upcoming years. Injury prevention and rehabilitation is one of the application domains of sport psychology where there is a considerable gap between research evidence and practice [92]. What is needed are well-controlled studies demonstrating the benefits of sport psychological interventions for both athletes and coaches alike.

One possible way to accomplish this goal is by taking into account the variability in individual athlete responses to psychological interventions. Ultimately, this strategy may lead to personalized treatment protocols and assessments. Models for the implementation of such tailoring of interventions to individual needs already exist in other fields such as medicine [93,94] and education (e.g., [95,96]).

In addition, for real progress in the field it will be important to take a proactive approach insofar as to consider the context in which the sport psychological protocols will be implemented. Specifically, this means taking into account the environments, available resources, and the needs and competencies of the professionals working with the athletes when designing and evaluating novel psychological measures and interventions. One example of such a novel and effective intervention protocol is storytelling [66].

In sum, the already existing and potential future possibilities in terms of sport psychological measures and interventions should allow for the specific selection and tailoring of these interventions to satisfy personal needs and preferences and to account for individual athlete responses to such interventions. Future developments in promising technologies such as artificial intelligence (e.g., [97,98]) and extended reality (i.e., augmented/mixed/virtual reality) (e.g., [99,100]) are likely to facilitate this endeavor.

As already mentioned before, a relatively new and promising approach to prepare athletes for a return to sport is through training with virtual reality (VR) [71]. A review article by Bird [70] provides a comprehensive overview of the possibilities and limitations of this technology. Numerous studies, including studies with junior athletes, have shown that VR training can improve sports-related skills without physical load. In addition, there is outstanding potential of this technology in terms of injury prevention and rehabilitation, as it enables athletes to engage in the learning, practice, and rehearsal of sports activities that in real life can be physically demanding or dangerous [71].

For instance, Pagé et al. [101] conducted an experiment in which junior basketball players viewed 50 short videos of set plays using VR goggles for five consecutive days. After each video, they were required to decide which action to perform next (e.g., pass, throw). As a result of the training intervention, the VR group demonstrated significantly better decision-making in both VR and on the court compared to another group who viewed the same videos on a regular screen. The most notable impact was observed in untrained plays, indicating that VR could enhance implicit and difficult-to-target skills such as pattern recognition, creativity, and game intelligence, after only one week of training. Therefore, junior athletes can develop their capabilities during injury rehabilitation and sharpen their abilities in the sport, irrespective of current physical limitations.

In addition to VR training, social connection exercises are an effective approach to improving mental aptitudes in sports and everyday life. These exercises (e.g., in the form of storytelling, as suggested by Arvinen-Barrow et al. [66]) aim to assist junior athletes in articulating their perspectives on their situations to others such as friends, training partners, and family members. This transparent communication can help junior athletes become more self-accepting and aware of their situations, feelings, and thoughts. Relationships and community are fundamental components of our lives, mental health, and mental performance. During a busy season, competition preparation, or a period of injury, however, junior athletes may neglect these important aspects.

Objective feedback from others—especially from neutral individuals—can assist in identifying emotional states and creating new perspectives in junior athletes. For instance, junior athletes may confide in a friend or family member about their apprehensions and aspirations for the ongoing or forthcoming rehabilitation phase. It is crucial to remain aware of one’s own emotions during conversation, as well as the reactions of the other individual. In addition, self-reflection of the subjectively experienced emotional response to the other person’s response is crucial for the athlete’s personal growth. Put differently, the ability of a junior athlete to be aware of their own reactions during an intimate conversation holds great potential for future psychological self-management [102].

## 4. Conclusions

Injuries remain a lurking danger in junior athletes’ daily lives. Sport psychology can contribute significantly to planning and executing effective and efficient prevention and rehabilitation measures for both elite and junior athletes. Specifically, stress-reduction methods, problem-situation diagnosis, and cognitive/mental skills training can aid athletes pre-, during, and post-injury, assisting with their psychological readiness to return to competition.

Sport psychology is a valuable tool for enhancing self-confidence and managing mental well-being in youth sports. Self-esteem and identity interventions aid junior athletes in accepting, embracing, and managing the realities of sports and life. This often results in not only improved mental health but also enhanced athletic performance. For this reason, actively engaging with measures from sport psychology represents an integral part on the athletic and personal path to excellence that should be utilized by all involved in youth sports.

## Data Availability

Not applicable.

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
