# Peer review of "The Role of Sport Psychology in Injury Prevention and Rehabilitation in Junior Athletes"

_behavsci, 2024, doi:10.3390/bs14030254_

Round 1

Reviewer 1 Report

Comments and Suggestions for Authors

The topic of the paper is interesting and important. However, there are some aspects in the present version that the authors should reconsider in order to improve the paper.

1) The focus of the paper is not clear enough. The aim of the paper, as stated in the abstract and in the paper itself, differs, which also affects the content of the paper - are the authors focusing on junior athletes or are they concerned with the whole community of athletes? In terms of sport psychology, this difference is important. The authors should clarify this point and rewrite the relevant parts of the paper, including expanding the list of references.

2. The title of the article is too ambitious, and the role of sport psychology in injury prevention and rehabilitation is not fully presented in the article. The authors should rewrite the title to accurately reflect the content of the article once they have moved beyond 1.

3. Much of the literature is out of date. Sports psychology is a rapidly growing field with many recent relevant contributions that should be adequately reflected in the article.

4. References are currently missing from the text, e.g. in lines 351-359.

Author Response

Response:

1) Thank you for this important comment. We now focus explicitly on junior athletes and make this clear throughout the text (and accordingly also changed the title of the manuscript; see next point).

Please see the clarifications in green font color throughout the manuscript.

2) We agree. We changed the title to better represent the content of out manuscript.

Please see line 3.

3) We substantially updated the literature and included a number of recent contributions to reflect the state-of-the-art in the field. In sum, 40 new studies are now included in the manuscript.

Please see the added references in green font color throughout the manuscript.

4) References for this paragraph and other sections are now included in the manuscript.

Please see lines 375-391 and beyond.

Reviewer 2 Report

Comments and Suggestions for Authors

We present a review work on the role of sports psychology in the prevention and recovery of sports injuries. We are faced with a document that is pleasant to read, which provides valuable information in an informative document, which can be considered as material for teaching, but which in my opinion does not provide the recent news on the subject that this work would require. Many of the topics raised are presented very superficially and with approaches that are not always current.

For example, out of 66 citations, only 8 are after 2019. They are not included in the same recent meta-analyses that would help to carry out a correct review (only two old meta-analyses are included, and they are indirectly linked to the content of the review). And the little recent information is that published by themselves in other magazines such as:

Richlan, F., Weiß, M., Kastner, P., & Braid, J. (2023). Virtual training, real effects: a narrative review on sports performance enhancement through interventions in virtual reality. Frontiers in psychology, 14, 1240790. https://doi.org/10.3389/fpsyg.2023.1240790

A journal of this category requires much more in-depth and current work on the state of the art.

Author Response

Response:

1) Recent news on the subject: We now provide a substantially more recent and in-depth discussion of current literature on the state-of-the-art in the field. We thank the reviewer for pointing out this important issue (see next point).

Please see the added references in green font color throughout the manuscript.

2) The inclusion of recent relevant literature is now reflected in the following numbers: 35 out of 106 citations are after 2019 (= 33%; compared to 12% in the previous version). We also included recent meta-analyses and systematic reviews on the topic.

Please see lines 37, 40, 43, 52, 149, 183, 308, 317, 336, 353, 369, 370, 378, 485, 486, 489, 496 for the added meta-analyses and systematic reviews.

Reviewer 3 Report

Comments and Suggestions for Authors

The article reviews the role of sports psychology in injury prevention and rehabilitation, emphasising the importance of psychological measures alongside physical approaches. It highlights the effectiveness of stress management techniques, cognitive restructuring, mindfulness, motor imagery, and seeking social support to prevent injuries and aiding athletes' return to competition. Next, we point out the major findings, positive and negative aspects, and finally possible improvements.

Major Findings:

-Psychological Interventions can significantly lower injury risk;

-The rehabilitation process benefits greatly from psychological support;

-Stress management, cognitive restructuring, and mindfulness are highly effective techniques.

Positive and negative aspects:

Positive

- Provides a comprehensive overview of psychological factors in sports injury prevention and rehabilitation.

-Highlights evidence-based psychological interventions.

Negative:

-The article could provide more detailed case studies or implementation examples regarding the study's theme.

-It might not sufficiently address the variability in individual athlete responses to psychological interventions.

Possible improvements for the future research:

-Authors could focus on tailoring interventions to individual needs;

-Development of novel psychological measures for a set of specific sports or injury types.

Author Response

Response:

1) Case studies or implementation examples: We now provide additional studies and examples.

Please see lines 207-212, 360-366.

2) Variability in individual athlete responses to psychological interventions: We added a paragraph on this issue and highlight this topic throughout the manuscript.

Please see lines 59, 207-212, 314-315, 370-373, 461-487.

3) Focus on tailoring interventions to individual needs: We now include a new section on future developments of interventions tailored to the individual needs of athletes.

Please see lines 62, 306, 337, 360-366, 370-373, 461-487.

4) Development of novel psychological measures: The new section also contains the development of novel psychological measures and interventions.

Please see lines 461-496.

Round 2

Reviewer 1 Report

Comments and Suggestions for Authors

The paper is improved. I recommend it for publishing.

Reviewer 2 Report

Comments and Suggestions for Authors

The paper has been substantially updated